# Hesperidin Ameliorates Sarcopenia through the Regulation of Inflammaging and the AKT/mTOR/FoxO3a Signaling Pathway in 22–26-Month-Old Mice

**DOI:** 10.3390/cells12152015

**Published:** 2023-08-07

**Authors:** Hyun-Ji Oh, Heegu Jin, Boo-Yong Lee

**Affiliations:** Department of Food Science and Biotechnology, College of Life Science, CHA University, Seongnam 13488, Gyeonggi, Republic of Korea; guswl264@naver.com (H.-J.O.); heegu94@hanmail.net (H.J.)

**Keywords:** hesperidin, sarcopenia, inflammaging, insulin growth factor-1, AKT/mammalian target of rapamycin/forkhead box 3a signaling pathway, aging

## Abstract

Faced with a globally aging society, the maintenance of health and quality of life in older people is very important. The age-related loss of muscle mass and strength, known as sarcopenia, severely reduces quality of life and increases the risks of various diseases. In this study, we investigated the inhibitory effect of hesperidin (HES) on inflammaging, with the intention of evaluating its potential use as a treatment for sarcopenia. We studied 22–26-month-old mice, corresponding to humans aged ≥70 years, with aging-related sarcopenia, and young mice aged 3–6 months. The daily administration of HES for 8 weeks resulted in greater muscle mass and strength and increased the fiber size of the old mice. HES also restored the immune homeostasis that had been disrupted by aging, such as the imbalance in M1/M2 macrophage ratio. In addition, we found that HES ameliorated the sarcopenia by regulating AKT/mammalian target of rapamycin/forkhead box 3a signaling through an increase in insulin-like growth factor (IGF)-1 expression in the old mice. Therefore, HES represents a promising candidate inhibitor of sarcopenia in older people, and its effects are achieved through the maintenance of immune homeostasis.

## 1. Introduction

Societies are aging, which increases the importance of maintaining a good quality of life and extending a healthy lifespan. Aging is a physiological process that is characterized by decreases in cellular lifespan, hormone secretion, and immune function [1]. This not only reduces the quality of life of older people, but also increases the risks of various diseases. The aging process also affects muscle tissue, causing reductions in the number of muscle fibers and the ability for muscles to grow [2,3]. Sarcopenia, which refers to the gradual loss of muscle mass and strength because of aging, has a significant impact on quality of life [4]. Sarcopenia is a degenerative disease, and the associated symptoms appear gradually. In the early phase, muscle loss and weakness develop, and, as it gradually worsens, restrictions to the activities of daily life and range of motion develop. In addition, sarcopenia can lead to psychological problems, such as anxiety, depression, and social isolation [5]. For these reasons, the amelioration of sarcopenia is considered to be important for the maintenance of health and quality of life, and for the prevention of disease in older people. Although the demand for improvements in age-related sarcopenia is increasing, no definitive means of management has been developed to date, and some strategies aimed at alleviating the symptoms of sarcopenia are not suitable for use in older people (e.g., high-intensity exercise or treatments for acute muscle damage) [6,7]. Therefore, there is an urgent need for fundamental and age-appropriate measures for the prevention and treatment of sarcopenia in older people.

Sarcopenia is the result of an imbalance in muscle protein metabolism secondary to aging [8]. When the balance between protein synthesis and breakdown within muscle is maintained, muscle mass remains constant [8,9]. However, with aging, muscle protein breakdown dominates over synthesis, leading to sarcopenia [9]. Therefore, an inhibition of muscle protein breakdown is considered to be the primary approach for the prevention of sarcopenia. It has been shown in previous studies that “inflammaging”, a low-grade chronic inflammatory state associated with aging, causes an imbalance in muscle protein metabolism and muscle loss [10,11,12]. Low-grade chronic inflammation leads to an inflammatory response and mitochondrial damage in muscle, which have negative effects on muscle function and growth [13,14]. In particular, macrophages, the principal immune cell type involved in this inflammation, can be categorized as M1 macrophages, which secrete pro-inflammatory cytokines, such as interleukin (IL)-6, IL-1β, and tumor necrosis factor (TNF)-α, which cause muscle loss, and M2 macrophages, which suppress excessive inflammatory responses and aid tissue recovery [15,16]. An increase in the number of M1 macrophages, which results in a disruption in the balance between M1 and M2 macrophages because of aging, leads to the excessive secretion of pro-inflammatory cytokines, which have direct effects on inducing the degradation of muscle proteins [17,18]. Therefore, the inhibition of inflammaging through the targeting of macrophages, while maintaining immune homeostasis, represents a key strategy for the treatment of sarcopenia in older people.

The forkhead box O (FoxO) signaling pathway promotes autophagy through its involvement in several cellular functions, including cell survival, proliferation, and metabolism [19,20]. The activation of FoxO in muscle cells leads to a higher expression of genes that encode proteins involved in muscle protein degradation [21]. FoxO3a, a member of the FoxO subclass, is a key regulator of muscle protein loss [22,23]. Sustained FoxO3a activity induces muscle loss by increasing the expression of the muscle-specific E3 ubiquitin ligases, F-box protein (Fbx32, also known as atrogin) and muscle ring-finger 1 (MuRF1) [22,23]. Thus, resistance to muscle wasting could be induced by targeting the inhibition of FoxO3a activity.

The restoration of a balanced muscle protein metabolism, by first inhibiting protein breakdown and then stimulating protein synthesis, should also be considered. Insulin-like growth factor (IGF)-1 is involved in cell growth and proliferation through the activation of the AKT/mammalian target of rapamycin (mTOR) signaling pathway, which stimulates protein synthesis [24,25]. In particular, the AKT/mTOR signaling pathway in muscle is important in muscle hypertrophy and atrophy [26]. AKT not only prevents FoxO3a from translocating to the nucleus and acting as a transcription factor, but also promotes protein synthesis by activating mTOR and its downstream substrates [27]. Age-associated reductions in sex hormone concentrations cause a reduction in the expression of IGF-1, the activation of AKT/FoxO3a signaling, and an inhibition of AKT/mTOR signaling [28]. However, it is necessary to confirm the importance of the AKT/mTOR/FoxO3a signaling pathway for the amelioration of sarcopenia, and also that of the expression of myogenic transcription factors, which are involved in myogenesis, such as myoblast determination protein 1 (MyoD), myocyte enhancer factor 2 (MEF-2), and myogenin [29].

Hesperidin (HES) is a flavanone glycoside that is abundant in the peel of fruits such as oranges and lemons [30,31]. Because HES has antioxidant, anticancer, and immune enhancing effects, it is considered to have various potential clinical applications [32,33,34]. In addition, since HES is a substance permitted as a food additive according to the Korean Ministry of Food and Drug Safety, it would be advisable to use it as functional food, given its proven healthy benefits. Although several beneficial effects have been demonstrated for HES, the effects of HES on sarcopenia have not been studied in detail. Therefore, in the present study, we evaluated the effects of HES on the sarcopenia of 22–26-month-old mice, corresponding to a human age of ≥70 years, and investigated the mechanisms involved. The overarching aim was to evaluate the potential for the use of HES for the treatment of sarcopenia in older people.

## 2. Materials and Methods

### 2.1. Reagents

The HES used in this study was obtained from Tokyo Chemical Industry (Tokyo, Japan), dissolved in distilled water, and administered orally at a dose of 5 or 10 mg/kg/day daily to mice.

For the flow cytometry analysis, FITC anti-mouse CD11b, AF647 anti-mouse F4/80, PE anti-mouse CD163, PerCP/Cyanine5.5 anti-mouse CD206, FITC anti-mouse CD45, and PE anti-mouse CD86 antibodies were purchased from BioLegend (San Diego, CA, USA). Specific antibodies against phospho-forkhead box O3a (p-FoxO3a), FoxO3a, phospho-AKT (p-AKT), AKT, phospho-mTOR (p-mTOR), mTOR, phospho-p70S6 kinase (p-p70S6K), and p70S6K were purchased from Cell Signaling Technology (Danvers, MA, USA). Antibodies against Fbx32, MuRF1, myostatin, and MyoD were purchased from Abcam (Cambridge, UK). Antibodies against phosphoinositide 3-kinase (PI3K), myogenin, and myocyte enhancer factor 2 (MEF-2) were purchased from Santa Cruz Biotechnology (Dallas, TX, USA).

### 2.2. Animals and Treatments

Female C57BL/6J mice aged 3–6 months or 20–24 months were purchased from the Korea Research Institute of Bioscience and Biotechnology (Daejeon, Republic of Korea). The animals were maintained in accordance with the standards specified in the “Guide to the Care and Use of Laboratory Animals” of the National Academy of Science, published by the National Institutes of Health. The animal experiments were approved by the Institutional Animal Care and Use Committee (IACUC) of CHA University (approval number IACUC220145). The mice were housed at 20 ± 3 °C under a 12 h light/dark cycle and were acclimatized to the facility for 1 week before being used in the study. The mice were randomly assigned to five groups (n = 10/group) as follows: (i) young control mice (aged 3–6 months; YM Ctrl), (ii) young mice administered 10 mg/kg/day HES (YM HES 10), (iii) old control mice (aged 20–24 months; OM Ctrl), (iv) old mice administered 5 mg/kg/day HES (OM HES 5), and (v) old mice administered 10 mg/kg/day HES (OM HES 10). Each concentration of HES was orally administered to the mice daily for 8 weeks. The body masses and muscle strengths of the mice were measured once a week. After the administration period, the mice were sacrificed and tissue samples were dissected, weighed, and stored at −80 °C until further analysis. The experimental protocol is summarized in Figure 1A.

### 2.3. Measurement of Grip Strength

The grip strengths of the mice were measured before the start of administration and once a week for the 8 weeks of the administration period. Limb grip strength was measured using a Chatillon force measurement system (Columbus Instruments, Columbus, OH, USA). The mice were placed on a mesh grid and allowed to grasp the mesh with all four feet, and then the tail of the mouse was gently pulled 3–5 times until it was about to fall off the grid. The force exerted at this moment was recorded as the mean grip strength of the mouse limbs.

### 2.4. Flow Cytometry Analysis

Spleen samples from the mice were minced, and single cells were obtained by passing the splenocytes through a 40 μm strainer. Red blood cells were dissolved in an ammonium–chloride–potassium lysis buffer (Lonza, Basel, Switzerland). After washing, single cells were stained with specific antibodies for 30 min on ice to allow for the count of the total number of macrophages (CD11b^+^F4/80^+^), M2a (CD11b^+^F4/80^+^CD163^−^CD206^+^), and M2c (CD11b^+^F4/80^+^CD163^+^CD206^+^) macrophages as well as the evaluation of the M0/M1 (CD11b^+^F4/80^+^CD163^−^CD206^−^) ratio.

Gastrocnemius (GAS) muscle samples were minced and incubated in collagenase and Dispase II (Sigma-Aldrich, St. Louis, MO, USA) at 37 °C for 1 h. The tissues were dissociated by passing the treated material through a 40 μm strainer. Single cells were stained with specific antibodies for 30 min on ice to permit the total number of macrophages (CD45^+^F4/80^+^) and the number of M1 (CD45^+^F4/80^+^CD86^+^CD206^−^) macrophages to be counted.

A flow cytometry was performed using a CytoFlex flow cytometer (Beckman Coulter, Brea, CA, USA) and the data were analyzed using FlowJo v10 software (Ashland, OR, USA).

### 2.5. Serum Analysis

After 8 weeks of treatment, blood samples were obtained from the mice by cardiac puncture at the time of sacrifice. The samples were centrifuged at 800× *g* for 15 min at 4 °C, and the serum samples obtained were stored at −80 °C until subsequent analysis. The serum concentrations of IL-6, IL-1β, and TNF-α were measured using a Mouse High-sensitivity T Cell Magnetic Bead Panel (Merck Millipore, Burlington, MA, USA). The serum concentrations of estradiol and IGF-1 were measured using a Mouse Estradiol ELISA Kit (MyBioSource, San Diego, CA, USA) and an IGF-1 ELISA kit (Thermo Fisher Scientific, Waltham, MA, USA), respectively. All these analyses were performed according to the manufacturers’ instructions. Absorbances were measured at appropriate wavelengths using a Luminex 100 analyzer (Luminex, Austin, TX, USA). All the sample concentrations were measured in triplicate.

### 2.6. Western Blotting

Samples of the quadriceps (QUA) and GAS muscles were finely diced and dissolved in Lysis buffer (iNtRON Biotechnology, Seoul, Korea) containing protease and phosphatase inhibitors for 30 min. The protein concentrations of the lysates were quantified using a BCA protein assay (Pierce, Rockford, IL, USA). Equal amounts of protein were subjected to SDS-PAGE and transferred to Immun-Blot PVDF membranes (Bio-Rad, Hercules, CA, USA). The membranes were blocked for 1 h in 5% skim milk and then washed with Tris-buffered saline containing 0.05% Tween-20, before being incubated overnight with the appropriate primary antibody at 4 °C. After washing the membranes again, they were blocked for 1 h in 5% skim milk containing the appropriate secondary antibody (peroxidase-conjugated anti-rabbit, anti-mouse, or anti-goat antibody, Bio-Rad). The protein signal intensity was measured using an EZ-Western Lumi Femto kit (DoGenBio, Seoul, Republic of Korea) and imaged using an LAS-4000 (GE Healthcare Life Sciences, Marlborough, MA, USA). The relative band intensities were quantified using ImageJ 1.48 software (NIH, Bethesda, MD, USA).

### 2.7. Histological Analysis

The QUA and GAS muscles were carefully separated from the hind legs of the mice, fixed in 4% paraformaldehyde, paraffin-embedded, and sectioned to generate 10 μm thick slices. The tissue sections were stained with hematoxylin and eosin for histological analysis. The sections were imaged using a Nikon E600 microscope (Nikon, Tokyo, Japan) and the cross-sectional areas (CSAs) of the muscle fibers were quantified using ImageJ software (NIH).

### 2.8. Statistical Analysis

All data are expressed as the mean ± standard error of the mean (SEM), and comparisons were made using one-way analysis of variance, followed by Tukey’s test. *p* < 0.05 was considered to represent statistical significance.

## 3. Results

### 3.1. HES Increases the Muscle Strength, Size, and Mass of the Old Mice and Preserves Body Mass

To evaluate the effects of HES on age-related sarcopenia, 22–24-month-old mice, which had progressive sarcopenia owing to natural aging, and which correspond to humans of ≥70 years, and 3–6-month-old mice, corresponding to 20–30-year-old humans, were studied [35]. The OM Ctrl group showed a steady decrease in body mass over the 8 weeks of the study. By contrast, in the OM HES5 and OM HES10 groups, body mass did not decrease throughout the 8 weeks of treatment with HES (Figure 1B).

Prior to the treatment, the grip strength of the OM Ctrl group was significantly lower than that of the YM Ctrl group, by more than two-fold. The grip strength of the OM Ctrl group then gradually decreased over the 8 weeks of the study, consistent with the progression of age-related sarcopenia, unlike in the YM Ctrl group. However, the 8-week treatment with HES not only prevented the decrease in grip strength, but also steadily increased the muscle strength of the OM HES5 and OM HES10 groups. In addition, the YM HES10 group showed a slight increase in grip strength (Figure 1C).

After sacrificing the mice, the sizes of the QUA and GAS muscles were compared. (Figure 1D), and we weighed them and normalized each mass to the final body mass of each mouse. We found that the masses of the QUA and GAS muscles of the OM Ctrl group were significantly lower than those in the YM group, but these were dose-dependently increased by 5 and 10 mg/kg/day HES (Figure 1E).

Thus, the weight loss accompanying the muscle loss and the reduction in muscle strength that occurred in the OM Ctrl group were suppressed by 8 weeks of HES administration, indicating that HES ameliorates sarcopenia in old mice.

### 3.2. HES Suppresses Inflammaging through the Regulation of Macrophage Subsets in Old Mice

Inflammaging, a term used to describe the low-grade chronic inflammation that occurs as part of the aging process, is closely related to muscle loss [18]. To build upon the previously demonstrated inhibitory effect of HES on sarcopenia, we investigated whether HES has a protective effect against inflammaging in old mice. The differences in macrophage subsets among the groups were identified by flow cytometry, using the following definitions: CD163^−^CD206^−^ cells as M0/M1, CD163^−^CD206^+^ cells as M2a, and CD163^+^CD206^+^ cells as M2c macrophages, based upon CD11b^+^F4/80^+^ macrophages identified in the spleen (Figure 2A). The percentage of CD11b^+^F4/80^+^ macrophages in the splenocytes was significantly greater in the OM Ctrl group than in the YM group; however, the increase was not present in the HES-treated OM group (Figure 2B,D). In addition, the percentage of M0/M1 macrophages was higher in the OM Ctrl group than in the YM Ctrl group, whereas those of M2c and M2a macrophages were low. HES treatment also tended to reduce or eliminate these differences in the populations of the M1, M2c, and M2a macrophages in the old mice (Figure 2C,E).

We next investigated whether HES affects the changes in the macrophages infiltrating GAS muscle tissue. The proportion of CD45^+^F4/80^+^ macrophages was higher in the OM Ctrl group than in the YM group, but lower in the OM HES5 and OM HES10 groups (Figure 3A,C). To characterize the changes in the ratio of M1 macrophages, which play a key role in inflammation in muscle tissue, CD86^+^CD206^−^ cells gated on CD45^+^F4/80^+^ macrophages were defined as M1 macrophages. As expected, the OM Ctrl group showed a much higher proportion of M1 macrophages than the YM group, but the OM HES-treated groups showed a similar proportion to the YM group (Figure 3B,D). The proportion of CD86^+^CD206^−^ cells, defined as M2 macrophages, showed no statistically significant difference between all groups. The greater abundance of macrophages in the OM Ctrl group, and especially that of M1 macrophages, is indicative of the development of inflammaging in the old mice. To further confirm this, the serum concentrations of pro-inflammatory cytokines that are secreted by macrophages were measured. We found that the serum concentrations of IL-6, IL-1β, and TNF-α, which are secreted by M1 macrophages, were much higher in the OM Ctrl group than in the YM group, but HES treatment reduced the concentrations of all three (Figure 3E). Taken together with the above findings, this suggests that HES ameliorates sarcopenia in old mice by ameliorating inflammaging, which causes muscle loss through effects on the macrophage population.

### 3.3. HES Reduces the Activation of FoxO3a and E3 Ubiquitin Ligases in Old Mice

To establish the molecular mechanism by which HES ameliorates sarcopenia, we measured the expression of FoxO3a, which causes muscle wasting by inducing proteolysis [23]. Unphosphorylated (i.e., activated) FoxO3a enters the nucleus and increases the expression of genes encoding proteins involved in the ubiquitin–proteasome system, whereas phosphorylated (i.e., not activated) FoxO3a remains in the cytoplasm and does not induce proteolysis [36]. The expression of p-FoxO3a/FoxO3a in the QUA and GAS muscles of the OM Ctrl group was lower than that of the YM group. This was associated with the higher expression of Fbx32 and MuRF1, which are E3 ubiquitin ligases activated by FoxO3a, in the OM Ctrl group. However, in the OM HES5 group, and especially in the OM HES10 group, the expression of p-FoxO3a/FoxO3a was significantly higher, and, accordingly, that of Fbx32 and MuRF1 was lower (Figure 4A,B). Taken together, these data imply that HES reduces muscle loss by inhibiting protein ubiquitination through the regulation of FoxO3a activation in the skeletal muscle of old mice.

### 3.4. HES Activates the AKT/mTOR Signaling Pathway in Old Mice

During aging, there is a close relationship between sex hormone concentrations and muscle mass. In particular, the decrease in estrogen concentration associated with aging results in increases in pro-inflammatory cytokine concentrations and a reduction in that of IGF-1, which induces protein synthesis, exacerbating sarcopenia [25,28]. Consistent with this, the serum concentrations of pro-inflammatory cytokines were higher (Figure 3E) and those of estradiol and IGF-1 were lower in the OM Ctrl group than in the YM group. However, the serum estradiol concentrations of the OM HES5 and OM HES10 groups were slightly increased by HES administration, and that of IGF-1 was also significantly increased (Figure 5A,B). Based on these results, we next investigated the effect of HES on the activation of the AKT/mTOR signaling pathway, which stimulates protein synthesis. The protein expression of PI3K, which is activated by IGF-1, was also higher in the QUA and GAS muscles of the OM HES-treated group. The p-AKT/AKT and p-mTOR/mTOR ratios of both muscles were lower in the OM Ctrl group but were significantly increased by 5 and 10 mg/kg/day of HES. The low level of activation of mTOR in the OM Ctrl group was associated with a low level of phosphorylation of p70S6K, a downstream signaling intermediate, but the changes in the ratio of p-p70S6K/p70S6K caused by HES administration were similar to those in the upstream intermediates in both muscles (Figure 5C,D). These results indicate that HES treatment increases the secretion of IGF-1 and promotes AKT/mTOR signaling, which corrects the deficiency in protein synthesis in the skeletal muscle of old mice.

### 3.5. HES Restores Muscle Fiber Size and the Expression of Myogenic Transcription Factors in Old Mice

To determine the effect of HES on muscle fiber size, we histologically analyzed the QUA and GAS muscles. In both muscles, the fibers were much smaller in the OM Ctrl group than in the YM Ctrl group, but the treatment of old mice with HES prevented the decrease in muscle fiber size, resulting in a significant recovery of the mean muscle fiber size in the OM HES5 and OM HES10 groups (Figure 6A,B). We also measured the expression of myogenic transcription factors in the QUA and GAS muscles of each group to determine whether HES upregulates myogenesis. We found that the expression of MyoD, MEF-2, and myogenin in the OM Ctrl group was lower than that in the YM Ctrl group in both muscles. However, HES administration increased the expression of all these proteins to the levels of the YM group (Figure 6C,D). These findings indicate that the administration of HES to old mice ameliorates the age-induced reduction in muscle fiber size and the inhibition of myogenesis.

## 4. Discussion

Sarcopenia is a disease that renders the activities of daily life difficult, owing to decreases in muscle mass and strength, and limits the health and quality of life of older people. The size and strength of muscles gradually decreases after the age of 30, and then rapidly decreases after the age of 70 [2]. Based on this, we studied mice aged 3–6 months, corresponding to ~30-year-old humans, and mice aged ≥22 months, corresponding to humans of ≥70 years. To use a mouse model of sarcopenia owing to natural aging, without acute muscle damage, the old mice used in the study were not chemically or physically treated. As expected, we found that all the parameters related to muscle mass and strength in the old mice were lower than in the young mice, confirming the establishment of a model of sarcopenia. However, the administration of HES for 8 weeks increased the muscle strength of the old mice and caused the maintenance of body mass. HES also increased the size and mass of the QUA and GAS muscles, which are the large muscles of the hind limbs, in the old mice. These results show the potential for HES to be used to ameliorate sarcopenia. Next, we aimed to explore the mechanisms involved.

The imbalance induced in the immune system by aging leads to the development of low-grade chronic inflammation, known as inflammaging, which contributes to the physiological dysfunction associated with aging by inducing damage to cellular components [37]. Furthermore, it has been shown in previous studies that inflammaging causes muscle loss [18,38,39]. As part of inflammaging, an imbalance between pro-inflammatory M1 macrophages and tissue repair-oriented M2 macrophages develops, and we confirmed the greater abundance of M1 macrophages and the lower abundance of M2 macrophages in the old mice using a flow cytometry analysis. However, HES administration was effective in preserving immune homeostasis by regulating the M1 and M2 macrophage populations. Pro-inflammatory cytokines, which are principally secreted by M1 macrophages, promote the degradation of myofibrillar proteins and reduce protein synthesis, thereby directly causing muscle wasting [40]. Consistent with its effects on the larger M1 macrophage population in old mice, the high serum concentrations of pro-inflammatory cytokines were also reduced by HES. These results indicate that the treatment of old mice with HES removes the principal cause of sarcopenia by restoring immune homeostasis by affecting the macrophage populations and inhibiting inflammaging.

Because the early phase of sarcopenia is characterized by the rate of muscle protein degradation exceeding the rate of synthesis, a realistic way of delaying sarcopenia in older people is to prevent muscle protein degradation. To determine whether HES inhibits protein degradation in muscle, we investigated the effect of HES on the expression of FoxO3a, a key regulator of proteolysis, and found that it inhibits FoxO3a activity, thereby preventing its nuclear translocation, in old mice. The inactivation of FoxO3a prevents its activity as a transcription factor, and, therefore, reduces the expression of the E3 ubiquitin ligases Fbx32 and MuRF1 [23,41], and we found that HES also blocked ubiquitin-proteasome signaling mediated by FoxO3 in old mice.

According to the results of previous studies, the age-associated decrease in estrogen concentration has a negative effect on muscle growth by increasing the circulating concentrations of inflammatory cytokines and reducing the activity of IGF-1 [28,42]. In particular, the aging-induced decline in IGF-1 accounts for the reduced PI3K activity, which in turn affects sarcopenia. IGF-1 is well-known as one of the key factors regulating protein synthesis by influencing the PI3K and AKT signaling pathways [43,44]. Aging-related declines in IGF-1 and a decrease in muscle PI3K activity are thought to be closely related and are important factors determining the deficit in muscle protein synthesis. Consistent with this, we found that the serum concentrations of estrogen and IGF-1 were lower in the old mice than in the young mice, and HES slightly ameliorated both changes. Based on this, we determined the effects of HES on the AKT signaling pathway, a key regulator of protein synthesis, and found that the phosphorylation of AKT was lower in the muscles from the old mice than in those from the young mice, which implies the existence of an impairment in the AKT signaling pathway during age-induced skeletal muscle loss. The lower AKT activity was associated with lower activities of the AKT targets mTOR and S6K1, which regulate cell growth and proliferation. However, the administration of HES restored the activities of AKT and mTOR in both muscles of the old mice. These results imply that HES stimulates muscle protein synthesis by restoring the activation of the AKT/mTOR signaling pathway in mice with sarcopenia. In addition, AKT activation inhibits muscle degradation by reducing the nuclear translocation of FoxO3a [36]. Taken together with the above findings, we have revealed that HES likely inhibits muscle loss by reducing muscle protein degradation and increasing synthesis through effects on the AKT/mTOR/FoxO3a signaling pathway.

The impairment of myogenesis that occurs with skeletal muscle aging contributes to sarcopenia [29,45]. The expression levels of MyoD and MEF-2, which are primary myogenic transcription factors, and myogenin, which is a secondary myogenic transcription factor, were much lower in muscles from old mice than in those from young mice. This is consistent with the smaller size of the muscle fibers in the old mice. However, HES treatment caused an increase in the expression of myogenic transcription factors in the old mice and a restoration of muscle fiber size. Thus, we have at least in part explained the effect of HES in ameliorating sarcopenia by demonstrating its effects on regulators of muscle protein synthesis and function.

In conclusion, we have shown that HES suppresses inflammaging, a major cause of sarcopenia, through the regulation of M1 macrophages in old mice. In addition, we found that HES reduces muscle protein degradation through the AKT/FoxO3a signaling pathway and induces muscle protein synthesis and myogenesis through AKT/mTOR signaling. Through these effects, the administration of HES to mice aged ≥22 months for 8 weeks caused improvements in muscle strength, mass, and fiber size. As expected, 8 weeks of treatment with HES, which is licensed as a food additive, did not show any negative effects on mice. Therefore, we suggest that HES can be used as a functional food for delaying sarcopenia in the elderly.

## Figures and Tables

**Figure 1 cells-12-02015-f001:**
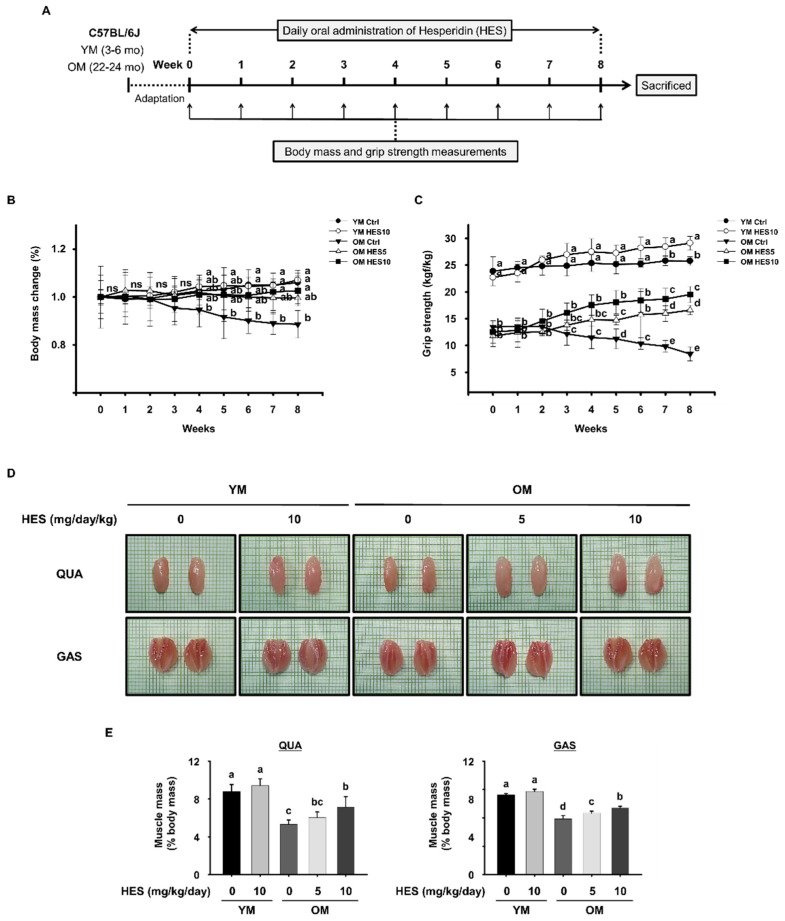
Effects of hesperidin (HES) on the body mass and muscle strength, size, and mass of old mice. (**A**) Experimental protocol: after a period of adaptation, young mice (3–6 months, YM) and old mice (22–24 months, OM) were orally administered HES (5 or 10 mg/kg/day) for 8 weeks. Afterwards, the mice were sacrificed for analysis. (**B**) Body mass change (%) and (**C**) grip strength, measured once a week for 8 weeks. (**D**) Representative images of the quadriceps (QUA, **top**) and gastrocnemius (GAS, **bottom**). (**E**) Masses of the QUA (**left**) and GAS (**right**) muscles, standardized to the body mass prior to sacrifice. “ns” indicates that there was no statistically significant difference between all groups. Different letters indicate that there were statistically significant differences; *p* < 0.05, a > b > c > d > e.

**Figure 2 cells-12-02015-f002:**
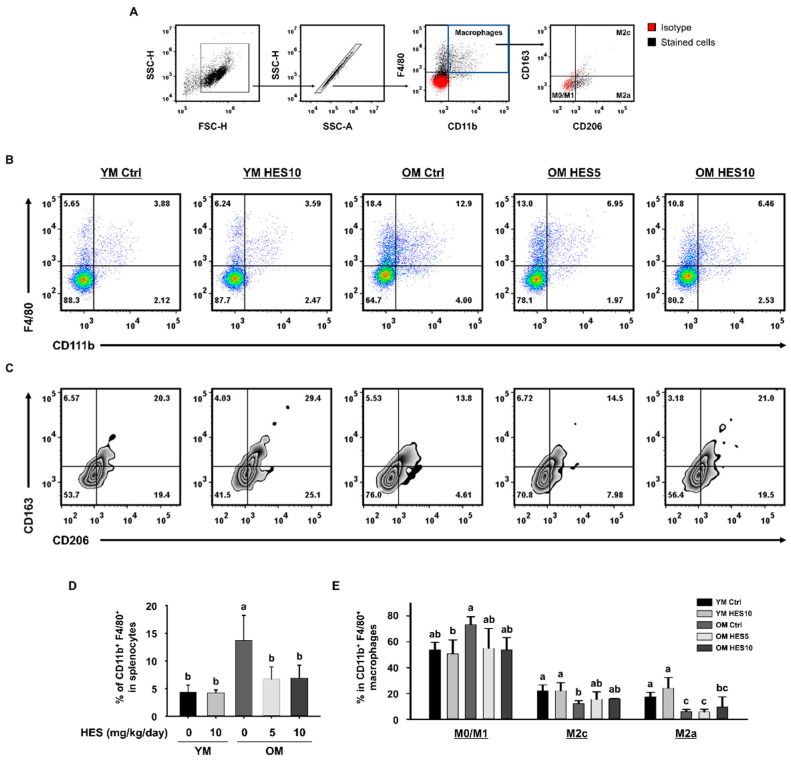
Effects of hesperidin (HES) on macrophage subpopulations in old mice. (**A**) Gating strategy for the analysis of macrophage subsets. Red dots indicate isotype controls and black dots indicate cells stained with specific antibodies. (**B**) Representative dot plots of CD11b^+^F4/80^+^ cells and (**C**) CD163^−^CD206^−^ M0/M1, CD163^−^CD206^+^ M2a, and CD163^+^CD206^+^ M2c macrophages. (**D**) Bar graph showing the mean % of CD11b^+^F4/80^+^ in the splenocytes. (**E**) Bar graph showing the mean % of M0/M1, M2c, and M2a macrophages among CD11b^+^F4/80^+^ cells. Different letters indicate statistically significant differences; *p* < 0.05, a > b > c.

**Figure 3 cells-12-02015-f003:**
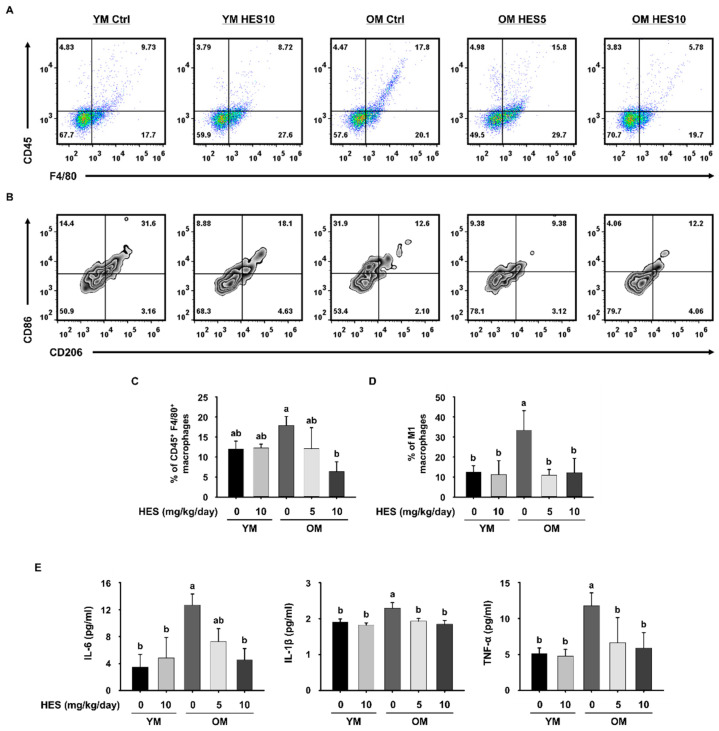
Effect of hesperidin (HES) on the M1 macrophage population and the serum concentrations of pro-inflammatory cytokines in old mice. Representative dot plots of (**A**) CD45^+^F4/80^+^ macrophages and (**B**) CD86^+^CD206^−^ M1 macrophages. (**C**) Bar graph showing the mean % of CD45^+^F4/80^+^ macrophages in gastrocnemius muscle. (**D**) Bar graph showing the mean % of M1 macrophages among CD45^+^F4/80^+^ cells. (**E**) Serum concentrations of pro-inflammatory cytokines. Different letters indicate statistically significant differences; *p* < 0.05, a > b.

**Figure 4 cells-12-02015-f004:**
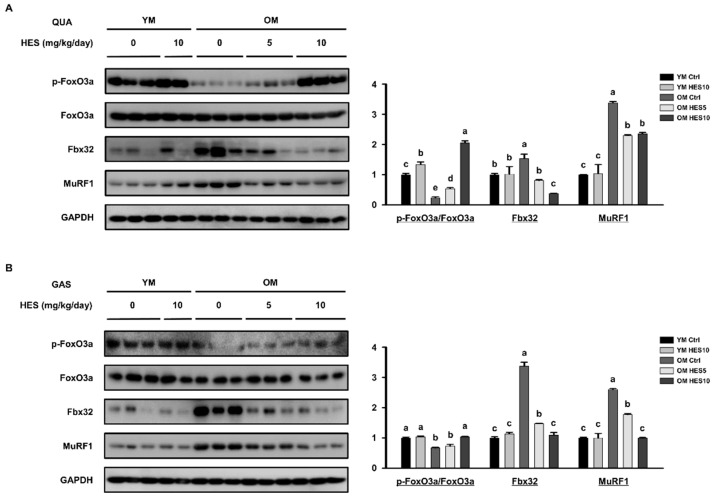
Effects of hesperidin (HES) on the expression of FoxO3a and E3 ubiquitin ligases in old mice. The protein expression of p-FoxO3a, FoxO3a, and E3 ubiquitination-related factors was measured by Western blotting in (**A**) QUA and (**B**) GAS muscles. GAPDH was used as a loading control and the phosphorylation of FoxO3a was normalized to the total expression of the protein. Different letters indicate statistically significant differences; *p* < 0.05, a > b > c > d > e.

**Figure 5 cells-12-02015-f005:**
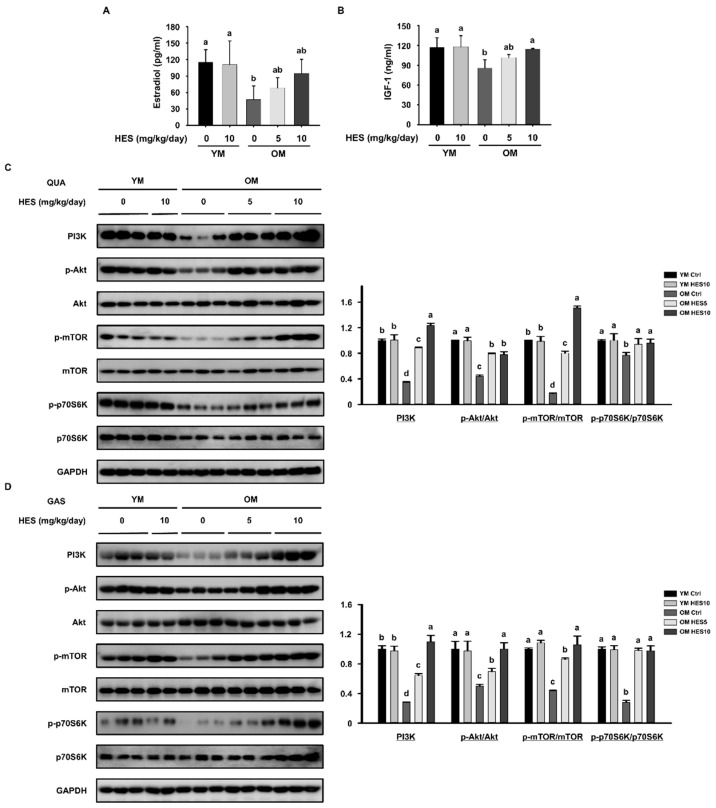
Effect of hesperidin (HES) on the AKT/mTOR signaling pathway in old mice. Serum concentrations of (**A**) estradiol and (**B**) IGF-1. The protein expression of components of the AKT/mTOR signaling pathway was measured using Western blotting in the (**C**) QUA and (**D**) GAS muscles. GAPDH was used as a loading control and the phosphorylation of each intermediate was normalized to the total expression level of each. Different letters indicate statistically significant differences; *p* < 0.05, a > b > c > d.

**Figure 6 cells-12-02015-f006:**
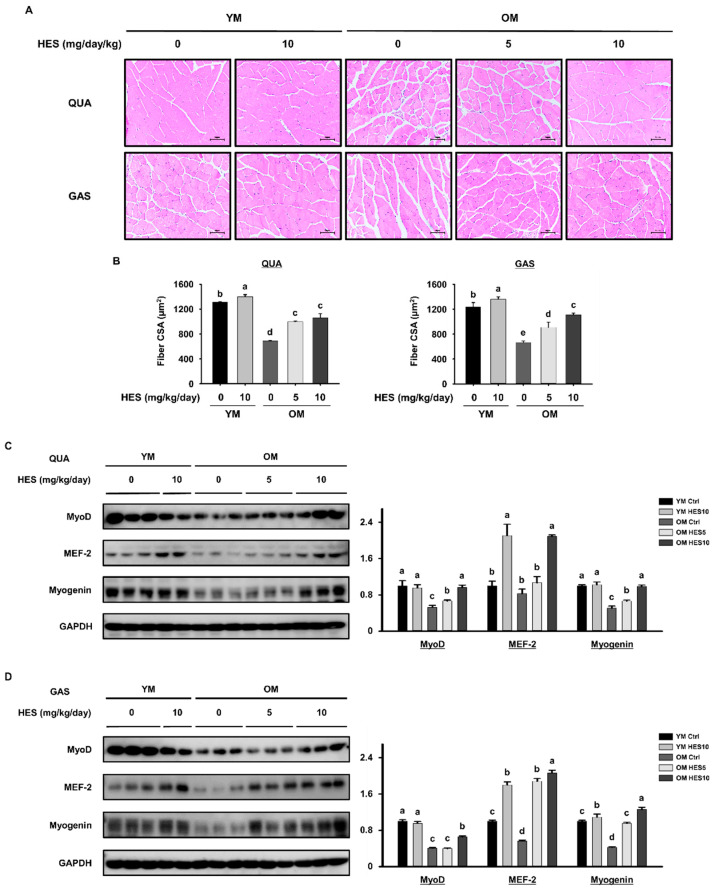
Effects of HES on muscle fiber size and the expression of myogenic transcription factors in old mice. (**A**) Histological analysis of hematoxylin- and eosin-stained QUA (**top**) and GAS (**bottom**) muscles. (**B**) Cross-sectional areas (CSAs) of the QUA (**left**) and GAS (**right**) muscles. The protein expression of myogenic transcription factors was measured by Western blotting in the (**C**) QUA and (**D**) GAS muscles. GAPDH was used as a loading control. Different letters indicate statistically significant differences; *p* < 0.05, a > b > c > d > e.

## Data Availability

All data generated and analyzed during this study are included in this article.

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
