# Peer review of "Hesperidin Ameliorates Sarcopenia through the Regulation of Inflammaging and the AKT/mTOR/FoxO3a Signaling Pathway in 22–26-Month-Old Mice"

_cells, 2023, doi:10.3390/cells12152015_

Round 1

Reviewer 1 Report

In this manuscript, titled “Hesperidin Ameliorates Sarcopenia through the Regulation of Inflammaging and the AKT/mTOR/FoxO3a Signaling Pathway in 22-26-Month-Old Mice”, the authors assess the contribution of two concentrations of hesperidin to ameliorate sarcopenia in old female mice compared to young female mice. The authors focus a large amount of their discussion on the role hesperidin on normalizing age-associated increases in inflammation. However, there are several flaws with both the presentation and the data collection in the current form of the manuscript, which significantly limits these findings. Other metrics involve associating hesperidin towards signaling pathways involved in muscle protein synthesis and degradation. Overall, this is an interesting paper, although improvements are needed in the macrophage characterization.

Major Concerns

11.      The authors state that they used Student’s t-test to calculate significance for the body mass data (Fig 1B). However, this appears to be a multivariate comparison, which is not appropriate for Student’s t-test, at least without Bonferonni or another suitable correction. The authors need to reanalyze the data with a one-way ANOVA, as they did for all other samples.

22.      The authors classify M1 macrophages as CD11b+F4/80+CD163-CD206+, and other notations for M2a and M2c (Section 2.4, lines 142-144), which are not conventionally correct, and then use more appropriate definitions (M1 as CD11b+F4/80+CD163-CD206-, and similar for M2a and M2c) in Section 3.2, lines 235-237. These definitions need to be uniform, and correct.

33.      In addition to point (2), the authors incorrectly label M1 as CD11b+ F4/80+. This definition is of inactivated macrophages (i.e., “M0”). To definitely label M1, another antibody, such as CD86, iNOS, or others, must be used.

44.      The text also refers to there being “more numerous” amounts of macrophages in the older populations than younger populations from analysis in the spleen (lines 237-238), and muscle (lines 252-254), therefore the total number of events in the flow cytometry should be provided to support this conclusion.

55.      The authors should discuss the quantity of M2 macrophages in the muscle, rather than just stating the M1 content, as the representative panels in Fig 3B seem to suggest HES treatment also increasing this population. Further, it is puzzling why the antibody panel changed at this point in the study (although, notably this panel includes a bona fide M1 marker). For consistency, results from the same panel should be provided throughout.

66.      PI3K is a common target in multiple signal transduction cascades. The conclusion that modest (albeit, statistically significant) downregulation of IGF1 would be the only reason for there to be a significant reduction in PI3K seems unlikely. The authors should adjust their discussion to reflect this.

77.      Sarcopenia is also characterized by a diminished presence of satellite cells (Pax7+) cells in muscle. While the results from MyoD and myogenin are promising, have the authors considered probing with Pax7? This would support and enhance these findings.

88.      Are there any potential negative consequences of long-term exposure to HES? The authors should add this to their discussion.

99.      The authors’ assertions that HES ameliorates sarcopenia in old mice seems somewhat overstated.

Minor Concerns

11.      The statistics in Figure 2E, Fig 3 C,E are confusing. The text seems to imply that “a” and “ab” are statistically significant, while the convention using this format usually has separate letters indicating significance (i.e. “a” is significantly different than “b”, but “ab” is not statistically distinct). These need to be revised.

22.      The y axis in Fig 2D is misleading. This is not the % of CD11b+F4/80+ macrophages, but the percentage of cells dissociated from the spleen.

33.      The authors could make it clearer that phosphorylated FoxO3a is the inactive and therefore “desirable” state. The current text is somewhat confusingly written (Section 3.3, lines 278-280)

English language easily readable with only minor edits needed.

Author Response

Thank you for your thoughtful review. We responded to each of your comments and revised our manuscript. Corrections were marked in red in "Revised manuscript". We hope that revisions based on your comments have further developed our manuscript. Thank you for your consideration. 

<Response to the major concerns>

1. Thanks for your comment. Agreeing with your opinion, we reanalyzed the body mass data (Figure 1B) and comparisons were made using one-way analysis of variance, followed by Tukey’s test. P < 0.05 was considered to represent statistical significance. We edited Figure 1B and corrected the sentences in line 190-191, 198-200, 224-226.

Revised line 190-191 : “All data are expressed as the mean ± standard error of the mean (SEM), and comparisons were made using one-way analysis of variance, followed by Tukey’s test.”

Revised line 198-200 : “The OM Ctrl group showed a steady decrease in body mass over the 8 weeks of the study. By contrast, in the OM HES5 and OM HES10 groups, body mass was maintained similar to week 0 throughout 8 weeks (Figure 1B).”

Revised line 224-226 : “ns” indicates that there was no statistically difference between all groups. Different letters indicate that there were statistically significant differences; p < 0.05, a > b > c > d > e.”

Revised Figure 1B : Please see the attachment.

2. There is an error in the definition of M1 macrophage on line 144. The mislabeled macrophage definition on line 144 was corrected same as in section 3.2.

Revised line 144 : “and the numbers of M0/M1 (CD11b+F4/80+CD163-CD206-)”

3. Thanks for your constructive comment. Indeed, we established the definition of macrophages based on several references, including [1, 2]. We judge that the definition of M1 we used (CD11b+F4/80+CD163-CD206-) is not entirely incorrect. However, considering the validity of your opinion, we acknowledged that our definition of M1 might include “M0”. So we decided to amend “M1” to “M0/M1” for clarity. We edited Figure 2A, E and related sentences.

Revised line 144 : “and the numbers of M0/M1 (CD11b+F4/80+CD163-CD206-)”, line 233 : “CD163CD206cells as M0/M1”,  line 237-239 : “In addition, the percentage of M0/M1 macrophages was higher in the OM Ctrl group than in the YM Ctrl group, whereas that of M2c and M2a macrophages were low.”,  line 246 : “(C) CD163CD206M0/M1”, line 248 : “% of M0/M1”

Revised Figure 2: Please see the attachment.

 [1] Lee, C.; Jeong, H.; Lee, H.; Hong, M.; Park, S. Y.; Bae, H., Magnolol Attenuates Cisplatin-Induced Muscle Wasting by M2c Macrophage Activation. Front Immunol 2020, 11, 77.

 [2] Zhu, Y.; Zhang, L.; Lu, Q.; Gao, Y.; Cai, Y.; Sui, A.; Su, T.; Shen, X.; Xie, B., Identification of different macrophage subpopulations with distinct activities in a mouse model of oxygen-induced retinopathy. Int J Mol Med 2017, 40, (2), 281-292.

4. According to your comment, we corrected the wording of the flow cytometry results. 

Revised line 235-239 : “The percentage of CD11b+F4/80+ macrophages in the splenocytes was significantly greater in the OM Ctrl group than in the YM group, but less so in the HES-treated OM group (Figure 2B, D). In addition, the percentage of M0/M1 macrophages was higher in the OM Ctrl group than in the YM Ctrl group, whereas that of M2c and M2a macro-phages were low.”

Revised line 252-255 : “We next investigated whether HES affects the changes in the pro-inflammatory M1 macrophages infiltrating GAS muscle tissue. The proportion of CD45+F4/80+ macrophages was higher in the OM Ctrl group than in the YM group, but lower in the OM HES5 and OM HES10 groups (Figure 3A, C).”

Revised line 257-259 : “As expected, the OM Ctrl group showed a much higher proportion of M1 macrophages than the YM group, but the OM HES-treated groups showed a similar proportion to the YM group (Figure 3B, D).”

5. First of all, we desired to investigate the changes in the proportion of overall macrophage subtypes. So, in order to see a more comprehensive distribution, the antibodies that distinguishes M1, M2a, M2c were used. As a results, the OM Ctrl group showed an imbalance of macrophages compared to the YM group, and HES showed a recovery effect in the OM HES-treated group. After confirming the overall macrophage subtypes, we aimed to observe changes in the infiltration of pro-inflammatory M1 macrophages into muscle , which are highly related to sarcopenia, in more specifically. Indeed, results of flow cytometry analysis showed that HES contributed to a decreased in the proportion of M1 macrophages in the GAS muscles of old mice. Along with serum analysis results, this suggests that HES ameliorates sarcopenia in old mice by improving inflammaging, which causes muscle loss through effects on the M1 macrophage population.

 In the same vein as above, we not only focused on pro-inflammatory M1 macrophage associated with sarcopenia, but also found that changes in the proportion of M2 macrophages in GAS muscles did not reveal statistically significant differences (Fig.3B is just one representative dot plot of the whole). Therefore, we did not discuss the quantity of M2 macrophages in the muscle (The data also not shown).

6. From your point of view, we agree with PI3K is a common target in multisignal transduction cascades. Aging-induced decline in IGF-1 may not be the only reason for the decline in PI3K activity, but it is a significant and influential reason in terms of sarcopenia. IGF-1 is well known to be one of the key factors in regulating the protein synthesis by influencing PI3K and its downstream signaling pathway. The decrease in IGF-1 due to aging and the decrease in PI3K activity in muscle are considered to be closely related, which is an important current in understanding deficits in muscle protein synthesis. As reflect your opinion, a discussion of IGF-1 and PI3K was added to the Discussion section.

Discussion section: In line 389-394 “In particular, aging-induced decline in IGF-1 is an important reason for reduced PI3K activity in terms of sarcopenia. IGF-1 is well known as one of the key factors regulating protein synthesis by influencing the PI3K and AKT signaling pathways [43, 44]. Aging-related declines in IGF-1 and decreases in muscle PI3K activity are thought to be closely related, which is an important current in understanding deficits in muscle protein synthesis.”

7. Thanks for your comment. Since Pax7 plays an important role in myogenesis, investigating whether HES inhibits the decrease of satellite cells (Pax7+) in old mice will improve our results. However, unfortunately, we could not consider the effect of HES on Pax7. Since MyoD and Myogenin are representative myogenic regulators, the fact that HES showed a significant increase in the expression of these two was sufficient to support our results. Nonetheless, we’ll reflect your comment if further research is conducted on the effect of HES on myogenesis. 

8. Hesperidin is a substance permitted as a food additive according to the Koran Ministry of Food and Drug Safety. Therefore, there was no negative effect on the long-term exposure of HES. Added this to the Introduction and Discussion section.

Introduction section: In line 92-94 “In addition, since HES is a substance permitted as a food additive according to the Korean Ministry of Food and Drug Safety, it is highly valuable to be used as a material for health functional foods using the proven effect of HES.”

Discussion section: In line 424-427 “As expected, 8 weeks of treatment with HES, which is licensed as a food additive, did not show any negative effects on mice. Therefore, we suggest that HES can be used as a functional food for delaying sarcopenia in the elderly.”

9. As noted in the Discussion (in line 357-360), our results showed that HES administration for 8 weeks improved muscle strength, size and mass in old mice. We strictly compared the statistically significant difference of each group using one-way ANOVA and Tukey’s test with p<0.05, and based on this, we showed that HES definitely contributed to improving sarcopenia in old mice. We analyzed the overall biomarkers related to muscle improvement, which supported the sarcopenia improvement effect of HES. Taken together, our results show that HES has a promising potential to be used as a health functional food ingredient for improving sarcopenia in the elderly.

<Response to the minor concerns>

1. Based on your comment, we corrected the notation in the figure legends to avoid misunderstandings (Deleted “ab” and “bc”).

Revised Figure Legends

Line 250 : “p < 0.05, a > b > c.”

Line 276 : “p < 0.05, a > b > c.”

Line 324 : “p < 0.05, a > b > c > d.”

Line 345 : “p < 0.05, a > b > c > d > e.”

2. According to your comment, we modified the y axis in Figure 2D from “% of CD11b+F4/80+ macrophages” to “% of CD11b+F4/80+ in splenocytes”

Revised Figure 2 : Please see the attachment.

3. Based on your comment, for clarification we modified the below:

In line 280-283 : “Unphosphorylated (i.e., activated) FoxO3a enters the nucleus and increases the expression of genes encoding proteins involved in the ubiquitin-proteasome system, whereas phosphorylated (i.e., inactivated) FoxO3a remains in the cytoplasm and does not induce proteolysis [36].”

Reviewer 2 Report

This manuscript reports the effects of Hesperidin (Hes), a flavanone glycoside obtained from the peel of citrus fruits, in countering the age-induced sarcopenia in mice. Mice were either 2-3 month-old (young adults) or 22-24 month-old (corresponding to ~70 yrs-old humans). Treatment lasted 8 weeks. Authors evaluated body mass, grip strength, masses of the quadriceps and of gastrocnemius. They were all improved in old mice following Hes treatment. Moreover, a number of parameters related to inflammation were evaluated, in particular the serum level of inflammatory cytokines and the amount of M1 macrphages. These parameters were elevated in old mice and dose-dependently decreased following Hes treatment. Since sarcopenia is based on the increase of proteolytic events in muscle cells, FoxO3a and E3 ubiquitin ligases were evaluated. As expected, they were found to be increased in old mice and reduced following Hes treatment. Finally, they evaluated the serum estradiol concentrations, which affect IGF-1 synthesis; while a decrease was observed in old mice, there was an increase in Hes-treated mice. In turn, such increase  mediated the activation of the AKT/mTOR signaling pathway, which stimulates protein synthesis.

The experiments are well conceived and the results converge into the demonstration that Hes may counteract sarcopenia in mice. Discussion of data is carried out in a correct way, without overinterpretation.

The only thing I would change is the way the significant differences are represented in the histograms. The method chosen by the Authors is not easy to visualize. I definitely prefer the classical representation with horizontal bars. 

Author Response

We greatly appreciate your kind comments. We partially agree with your opinion, but we hope you understand that not all representations can be converted to horizontal bars. Those showing statistically significant differences were marked with different letters (a, ab, b, bc, c, cd, d) so that they could be well visualized, and this is considered to sufficiently represent the difference between groups.

Round 2

Reviewer 1 Report

In this revised manuscript, the authors were very responsive to previous comments. The manuscript reads well and my only minor comment would be to include a statement regarding the quantity of the M2 macrophages in the GAS muscles (the data do not necessarily need to be shown in detail). A similar presentation as in Fig 2E would be sufficient. This would complete the story of the characterization of splenocytes in circulation, and then the relative amount of macrophages in the GAS muscle.

Author Response

Thanks for your comment. Based on your comment, we added the following statement to the Result 3.2.

Added line 259-261 : “The proportion of CD86+CD206− cells, defined as M2 macrophages, showed no statistically significant difference between all groups (data not shown).”

Reviewer 2 Report

I maintain the previously expressed opinion regarding the way of representing statistical significance in histograms. The horizontal bars can be multiple and the significance is more immediately understandable than the use of small letters whose meaning must be sought in the caption. I don't think this flaw is such as to change my positive opinion of the manuscript, but I found the authors' response opinionated.

I also point out that many of the sentences added, probably to respond to another reviewer's requests, are to be reviewed as English. Below is the fix I suggest.

Lines 93-4. … it is highly valuable to be used as a material for health functional foods using the proven effect of HES. >>> it would be advisable its use as functional food, given its proven healthy benefits.

Lines 144-7. … to permit the total number of macrophages (CD11b+F4/80+ ), and the numbers of M0/M1 (CD11b+F4/80+CD163-CD206- ), M2a (CD11b+F4/80+CD163-CD206+ ), and M2c (CD11b+F4/80+CD163+CD206+ ) macrophages to be counted. >>> to allow for the count of the total number of macrophages (CD11b+F4/80+ ), M2a (CD11b+F4/80+CD163-CD206+ ), and M2c (CD11b+F4/80+CD163+CD206+ ) macrophages as well as the evaluation of the M0/M1 (CD11b+F4/80+CD163-CD206- ) ratio.

Lines 200-1. … body mass was maintained similar to week 0 throughout 8 weeks (Figure 1B). >>> body mass did not decrease throughout the 8 weeks of treatment with HES.

Line 237. … but less so in the HES-treated OM group (Figure 2B, D). >>> however, the increase was not present in the HES-treated OM group (Figure 2B, D).

Line 283. … (i.e., inactivated) FoxO3a >>> (i.e., not  activated) FoxO3a

Lines 391-5. … aging-induced decline in IGF-1 is an important reason for reduced PI3K activity in terms of sarcopenia. IGF-1 is well known as one of the key factors regulating protein synthesis by influencing the PI3K and AKT signaling pathways [43, 44]. Aging-related declines in IGF-1 and decreases in muscle PI3K activity are thought to be closely related, which is an important current in understanding deficits in muscle protein synthesis. >>> aging-induced decline in IGF-1 accounts for the reduced PI3K activity, which in turn affects sarcopenia….. Aging-related decline in IGF-1 and decrease in muscle PI3K activity are thought to be closely related, and are important factors determining the deficit in muscle protein synthesis.

The sentences added in the revision have some language issue. I included suggestion for fixing

Author Response

Thanks for your comment. Please understand that it is difficult to convert all representations to horizontal bars. We think that using captions (a, b, c, d..) to indicate statistical significance is one of the most common and easy-to-understand way to show differences between several groups.

And also, based on your suggestion, we modified the sentences as follows:

Lines 93-94 : “,it would be advisable its use as functional food, given its proven healthy benefits.”

Lines 143-146 : “to allow for the count of the total number of macrophages (CD11b+F4/80+), M2a (CD11b+F4/80+CD163-CD206+), and M2c (CD11b+F4/80+CD163+CD206+) macrophages as well as the evaluation of the M0/M1 (CD11b+F4/80+CD163-CD206-) ratio.”

Lines 199-200 : “By contrast, in the OM HES5 and OM HES10 groups, body mass did not decrease throughout the 8 weeks of treatment with HES (Figure 1B).”

Line 236-237 : “however, the increase was not present in the HES-treated OM group (Figure 2B, D).”

Line 282 : “(i.e., not activated) FoxO3a”

Line 390-394 : “, aging-induced decline in IGF-1 accounts for the reduced PI3K activity, which in turn affects sarcopenia. IGF-1 is well known as one of the key factors regulating protein synthesis by influencing the PI3K and AKT signaling pathways [43, 44]. Aging-related declines in IGF-1 and decrease in muscle PI3K activity are thought to be closely related, and are important factors determining the deficit in muscle protein synthesis.”

Thanks for your kind suggestion.